# Recent Advances in Proteomics-Based Approaches to Studying Age-Related Macular Degeneration: A Systematic Review

**DOI:** 10.3390/ijms232314759

**Published:** 2022-11-25

**Authors:** Laura García-Quintanilla, Lorena Rodríguez-Martínez, Enrique Bandín-Vilar, María Gil-Martínez, Miguel González-Barcia, Cristina Mondelo-García, Anxo Fernández-Ferreiro, Jesús Mateos

**Affiliations:** 1Pharmacology Group, Health Research Institute of Santiago de Compostela (FIDIS), 15706 Santiago de Compostela, Spain; 2Pharmacy Department, University Clinical Hospital of Santiago de Compostela (SERGAS), 15706 Santiago de Compostela, Spain; 3Ophthalmology Department, University Clinical Hospital of Santiago Compostela (SERGAS), 15706 Santiago de Compostela, Spain

**Keywords:** proteomics, age-related macular degeneration, inflammation, biomarker, oxidative stress

## Abstract

Age-related macular degeneration (AMD) is a common ocular disease characterized by degeneration of the central area of the retina in the elderly population. Progression and response to treatment are influenced by genetic and non-genetic factors. Proteomics is a powerful tool to study, at the molecular level, the mechanisms underlying the progression of the disease, to identify new therapeutic targets and to establish biomarkers to monitor progression and treatment effectiveness. In this work, we systematically review the use of proteomics-based approaches for the study of the molecular mechanisms underlying the development of AMD, as well as the progression of the disease and on-treatment patient monitoring. The Preferred Reporting Items for Systematic Reviews and Meta-Analysis (PRISMA) reporting guidelines were followed. Proteomic approaches have identified key players in the onset of the disease, such as complement components and proteins involved in lipid metabolism and oxidative stress, but also in the progression to advanced stages, including factors related to extracellular matrix integrity and angiogenesis. Although anti-vascular endothelial growth factor (anti-VEGF)-based therapy has been crucial in the treatment of neovascular AMD, it is necessary to deepen our understanding of the underlying disease mechanisms to move forward to next-generation therapies for later-stage forms of this multifactorial disease.

## 1. Introduction

AMD is the leading cause of blindness in the elderly population in Western countries [1], affecting nearly 300 million people worldwide who are visually impaired, either partially or totally. AMD is characterized by progressive degenerative and/or neovascular changes affecting the macula, the highly specialized region of the central retina responsible for fine vision [2]. AMD can be divided into three different stages: early, intermediate and advanced AMD. Regarding advanced AMD, it can be subdivided into geographic atrophic (GA) or “dry” AMD, which represent around 80% of cases [3], and the rapidly blinding neovascular form (nAMD), also called “wet” or “exudative” [4]. Whereas GA is characterized by initial RPE degeneration, in nAMD, loss or dysfunction of the choroidal vasculature is the first pathological event [5].

The best available treatment, based on anti-vascular endothelial growth factor (anti-VEGF) intra-vitreal injections, is useful exclusively in patients suffering from nAMD [6,7]. However, this treatment, in some cases, only delays the progression of the disease [8,9]. Furthermore, continuous anti-VEGF treatment has been linked to drug intolerance and occasional development of GA, thus compromising the long-term benefits to the patients [10].

From an etiologic point of view, AMD is a multifactorial disease, determined by genetic and non-genetic factors [11]. Aging appears to be the most critical factor since the prevalence of the disease progressively increases with older age [1]. A strong association has been described with mutations in genes such as the HTRA1/ARMS2 locus [12], and with complement components such as complement factor H (CFH) and complement C3 (C3) [13,14,15]. Finally, several environmental and systemic risk factors such as obesity [16], hypertension [17] and hypercholesterolemia [18,19] predispose people to the development of AMD.

At the molecular level, the pathogenesis of AMD is influenced by the generation of highly reactive free radicals in the macular area of the retina, a zone characterized by a high metabolic rate due, in part, to high oxygen pressure and redox reactions being continuously generated [20]. It is widely believed that the presence of reactive oxygen species (ROS) is strongly linked to the pathogenesis of AMD [21,22]. The combination of chronic oxidative stress, subsequent impaired autophagy and inflammation leads to the aging of Retinal Pigment Epithelial (RPE) cells [23]. In the dry form of AMD, the compromised capacity to neutralize mitochondrial-derived ROS and impaired proteostasis cause a detrimental accumulation of lysosomal lipofuscin, a complex non-degradable polymeric lipid–protein mix [24] that forms extracellular structures called “drusen” [25], localized between the basal lamina of the RPE and the inner collagenous layer of Bruch’s membrane (BrM). The formation of these deposits is one of the hallmarks of aging in the eye and their size and number predict the progression and the degree of the dry form of the disease [26]. Drusen formation activates the NACHT, LRR and PYD domain-containing protein 3 (NLRP3) inflammasome via the complement C1q (C1Q), a component of drusen [27]. NLRP3 is one of the central molecules involved in pyroptosis, a programmed cell death characterized by swift plasma membrane disruption and the subsequent release of cellular content, including typical pro-inflammatory mediators such us IL-1β and IL-18 [28]. Furthermore, changes in the extracellular matrix (ECM) affecting collagen layer structure and elasticity can promote the loss of differentiation and the epithelial mesenchymal transition (EMT) of healthy RPE cells [29,30]. It is accepted that interactions between the RPE layer and the fibrous, acellular BrM are critical in the pathogenesis of AMD [31]. Both structures form the blood–retinal barrier, involved in the maintenance of the health of the retina through the exchange of nutrients, oxygen and waste products with the choroid. On the contrary, in nAMD, retinal pigment epithelial displacement and damage occur because of choroidal neovascularization (CNV) through the BrM, leading to fluid accumulation [32]. This process is driven by angiogenic factors such as VEGF, whose expression is increased in RPE cells in nAMD [2]. The neovascularization process can be classified in three forms using optical coherence tomography (OCT): Type 1 CNV refers to vessels beneath the RPE, whereas type 2 CNV is characterized by vessels expanding into the subretinal space between the neurosensory retina and the RPE [33], and type 3 by retinal angiomatous proliferation [34].

A recent study revealed that early AMD signs can already be detected in patients under 30 years old [35]. Due to the absence of an effective preventive treatment, the number of patients severely disabled by AMD is expected to increase up to 50% in the coming decades [36]. The disease not only exerts a tremendous impact on the physical and mental health of the geriatric population and their kindred, but it is also becoming a major public health issue and financial burden. Thus, it is crucial to find strategies to identify patients at high risk of developing AMD and to improve their management.

The term proteomics encompasses all research methodologies developed for the qualitative and quantitative study of the proteome, which are the proteins present in a cell type, tissue or organism at a given stage of development [37]. In the last decade, there has been an exponential increase in the use of proteomics in translational research due, in large part, to the progress in state-of-the-art mass spectrometry, an analytical technique, developed in the middle of the last century [38]. Proteomics has emerged as a powerful tool for biomarker discovery [39], both independently and in combination with complementary, non-MS-based proteomic approaches such us antibody-based multiplex assays, multiplexed enzyme-linked immunosorbent assays (ELISA) and aptamer-based techniques [40,41]. As regards the specificity and reliability of not only diagnostic tests, but also of treatment targets, the source of biomarkers is of paramount importance. Non-invasiveness is a key factor for any diagnostic approach. Tear fluid, for instance, represents a precious source of biomarker panels for disease progression and response to treatment in AMD. First, it is the nearest biological fluid to the pathological spot, the posterior cavity of the eye [42]. Second, it is easily accessible and can be collected via minimally invasive methods. Third, the protein content is relatively high, ranging from 6 to 10 mg/mL [43]. Tear fluid has been revealed, in recent decades, as a source for biomarker discovery, since is an extremely complex biological mixture of proteins, lipids, metabolites and salts [44]. Up to 1500 different proteins can be identified via quantitative shotgun proteomics [45]. Although tear fluid offers numerous advantages, other fluids and tissues are, at least a priori, promising sources for biomarker discovery and for the study of the pathogenesis of AMD [46]. 

In a previous work from 2018 [47], Kersten et al., comprehensively reviewed the use of systemic and ocular fluids to identify compounds, including proteins but also metabolites, lipids, auto-antibodies and miRNAs, as potential biomarkers in AMD. However, to the best of our knowledge, there is a lack of systematic reviews specifically focused on proteomic studies. Thus, our aim for this review was to systematically compile the most relevant proteomics-based studies on AMD with a special focus on those covering, during the last five years, new biomarkers and therapeutic targets in GA.

## 2. Methods

### 2.1. Database Retrieval and Search Strategy

Electronic bibliographic databases including PubMed and Web of Science were used to search published research papers. The design of this study followed the guidelines of the Preferred Reporting Items for Systematic Reviews and Meta-Analysis Protocol (PRISMA-P) [48]. The search filter included the following terms combined with “AND”: “age-related macular degeneration” and “proteomics”. The published language was limited to English and the search results were screened for suitable topics and full articles accessible for systematic review. The workflow is summarized in Figure 1. Only human studies were included, whereas experimental methods and protocols, reviews, systematic reviews, preprinted articles and conference proceedings and abstracts were excluded.

### 2.2. Data Extraction

Three independent reviewers extracted data from each eligible study using a standardized data-extraction sheet. Later, the results were cross-checked and disagreements between both reviewers regarding the extracted data were resolved through discussion with a fourth reviewer. Thirty-seven published research articles were selected and grouped according to the biological source of biomarkers used for the proteomic study (Table 1, Table 2 and Table 3).

## 3. Results and Discussion

This section is structured in sub-sections according to the different types of human samples used for the proteomic study, starting from the closest structures to the macula and finishing with the systemic fluids. Furthermore, an initial sub-section summarizes the different proteomic approaches found in the literature.

### 3.1. Recent Advances in Proteomic Approaches to the Study of the Disease

Old-fashioned proteomic approaches such as Peptide Mass Fingerprinting (PMF) or Differential In-Gel Electrophoresis (DIGE) followed by Matrix-Assisted Laser Desorption/Ionization–Time-Of-Flight mass spectrometry (MALDI-TOF) identification are progressively being replaced by more modern high-resolution quantitative techniques such as Liquid Chromatography coupled on-line to mass spectrometry (LC-MS/MS), allowing for deeper identification and more robust quantitation. Historically, most of the quantitative shotgun proteomics-based studies have been conducted using data-dependent acquisition (DDA) methods, where the mass spectrometer settings are adjusted to isolate and fragment peptides based on the intensities observed in MS1 survey scans. Basically, the top (usually 10–20) most intense peptides for each time point along the chromatographic gradient are selected for fragmentation and subsequent MS2 level identification, so the resulting MS/MS spectra are assigned to specific peptide sequences via protein database matching. To avoid redundant acquisition due to fragmentation of the same precursor at consecutive time-points, dynamic exclusion can be applied, leading to higher protein coverage and the detection of low-abundance proteins [85]. Examples of DDA techniques include classical label-free, stable-isotope labeling by amino acids in cell culture (SILAC) and chemical-based labeling such as Tandem Mass Tags (TMT) or isobaric labeling (iTRAQ) [86,87,88]. DDA approaches usually lead to the identification of very complex sets of proteins. However, they have inherent drawbacks related to the stochastic nature of peptide ionization and fragmentation, such as lack of reproducibility/accuracy in the quantification [89].

During the last decade, the emergence of unbiased data-independent acquisition (DIA) methods has revolutionized the field, avoiding problems derived from the stochastic nature of the peptide ionization. Sequential Windowed Acquisition of All Theoretical Fragment Ion Mass Spectra (SWATH-MS) and Hyper Reaction Monitoring (HMR-MS) are highly robust and reproducible label-free techniques in which the mass spectrometer settings are adjusted to isolate and fragmentate all the precursors detected within slightly overlapping windows, covering the entire working m/z range across the entire chromatographic gradient [90]. In this case, the search is performed not using protein databases, but using spectral libraries previously generated via DDA of a pool of the samples instead [91]. Last-generation software packages include special algorithms capable of generating those spectral libraries on the fly using the same DIA data acquired in the studied samples [92,93], thus improving data processing speed and reproducibility.

In any case, protein extracts from cultured cells or tissues/biological fluids are highly complex samples that exhibit a wide dynamic range of concentrations [94]. Hence, for quantitative proteomics, it is generally necessary to quantify the samples, both at the level of total proteins and, subsequently, prior to injection in the LC-MS system, at the level of peptides. In DDA-based approaches, fractionation of the sample is often necessary to reduce its complexity and is therefore highly recommended. On the contrary, fractionation is not recommended for DIA approaches [89].

One of the main limitations of LC-MS-based proteomic techniques is the low sensitivity for identifying scarce proteins such as cytokines or growth factors, especially in complex samples or samples with a high dynamic range of protein content [95]. Alternatives to overcoming this limitation are classical antibody-based technologies such as multiplex techniques and ELISA, or more recent aptamer-based approaches, relying on single-stranded library DNA-based reagents with high binding specificity and complementarity to target proteins [96]. The reagents are immobilized and incubated with the protein sample to be tested. After washing and removing the unbound fraction, the protein–reagent complexes are again immobilized, and the DNA-based reagent is eluted and quantified using standard techniques.

### 3.2. Proteomics on Retinal Pigment Epithelial Cells and Extracellular Vesicles in AMD

The RPE constitutes a cell monolayer essential to maintaining normal photoreceptor function (Figure 2). RPE not only participates in the visual cycle, but also provides nutrients to the photoreceptors and is responsible for withdrawing waste debris from their outer segments [8]. Compromised molecular regulation between the RPE layer and the BrM is a hallmark of the early stage of AMD [97].

A combination of transcriptomics and proteomics was employed by Zauhar et al. [58], to dissect the role of multiple retinal and choroidal cell types (Müller glia, neurons and RPE/choroid) in determining complement homeostasis. The results indicated that this process has a key role in the involvement of RPE cells in progression to late AMD. Recently, another transcriptome and proteome-based study has identified pathways specifically modulated in GA. Induced pluripotent stem cells (iPSCs) were generated from fibroblasts from a cohort of 43 individuals with GA and 36 controls with genotype data [57]. In this work from Senabouth et al., mitochondrial dysfunction, and specifically, an increase in Complex I linked to a higher oxygen consumption rate, was identified as a central genetic factor associated with GA.

Extracellular vesicles (EVs) have been revealed as key players in biological processes such as aging, cell homeostasis and disease [98,99]. In fact, the molecular cargo of the EVs secreted by the RPE cells has an important role in the pathogenesis of AMD [100]. A non-quantitative proteomic approach was used to merely identify the proteins contained in the so-called plasma membrane blebs of ARPE-19 cells incubated with hydroquinone, a major component in cigarette smoke [50]. Glycosylated forms of basigin and MMP-14 were localized in those blebs and the authors proposed involvement of these proteins in extracellular matrix remodeling at sites distal to the RPE, potentially contributing to the progression of “dry” AMD. Biasutto et al. [53], by using reversed-phase protein assays, identified a subset of phosphorylated proteins that are characteristic of ARPE-19 cells cultured under oxidative stress conditions. Interestingly, some of those phosphorylated isoforms, such as PDGFRβ, VEGFR2 and c-kit, were also detected in the vitreous of AMD patients. Previously, C5b-9 had been identified as part of the coating of the EVs released by stressed RPC cells, suggesting a role of these vesicles in the focalized modulation of complement activation [51].

In a recent study, the pooled sera of both smokers (n = 32) and non-smokers (n = 35) were collected and used to treat RPE cells obtained from the eyes of four donors harboring high-risk ARMS2/HTRA1 alleles for AMD, and the eyes of two donors with low-risk alleles [56]. iTRAQ was used to identify differentially expressed proteins (DEPs). Under the effect of smokers’ serum, 464 DEPs were identified in the high-risk group (smokers vs. non-smokers). In contrast, in the low-risk group, the number of DEPs decreased to merely 30. Gene ontology analysis showed that smokers’ serum enhanced molecular pathways involved in Alzheimer’s disease, oxidative phosphorylation and RPE phagocytic function. Interestingly, caveolin-1 and HTRA1 were among the most significantly upregulated proteins in the high-risk group vs. the low-risk group after exposure to smokers’ serum, strongly supporting a gene–environment interaction between the high-risk alleles ARMS2/HTRA1 and smoking in the occurrence and development of AMD.

Recently, Flores-Bellver et al. [55] have taken a step forward in this field. These authors generated RPE cells from CD34+ cord blood mesenchymal stem cell (MSC)-derived iPSCs. The induced primary RPE cell monolayers presented hallmarks of cell differentiation and key physiological characteristics of the native RPE tissue, e.g., the expression of genes involved in essential RPE functions such as functional apical–basal polarization and EV secretion. The proteomic analysis showed that EVs contained proteins involved in AMD pathogenesis and drusen formation and revealed apical–basal directional proteome enrichment. The monolayers were then treated with increasing concentrations of cigarette smoke extract (CSE) to study the effects of both acute and chronic stress. An increase in drusen-related proteins was detected in the cargo of the EVs released under chronic stress conditions.

### 3.3. Proteomics on Bruch´s Membrane in AMD

BrM is a thin, stratified, extracellular matrix whose main physiological role is structural, but also facilitates transport to help regulate the diffusion of nutrients and waste products between the RPE and the bloodstream [101]. BrM undergoes significant age-related changes, including thickening and decreased permeability, that disrupt normal retinal physiology and contribute to AMD [102]. One of its five differentiated layers is composed mainly of elastin [103]. It has been hypothesized that degradation of elastin at the BrM macula is a key event facilitating CNV [29]. In line with this are the elevated serum levels of elastin-derived peptides (S-EDPs) [73] as well as elastin autoantibodies [104] found in nAMD patients vs. eAMD patients and healthy controls.

A quantitative proteomic analysis of BrM using iTRAQ-based chemical labeling was performed on post-mortem collected samples [52]. Nine hundred and one proteins were quantified. Most proteins did not differ in amount between the AMD and control samples, reflecting the normal proteome of an average 81-year-old individual. A total of 56 proteins were found to be overexpressed and about 60% of these, including α-defensins 1–3, histones and galectin-3, were involved in immune response and host defense, strongly supporting the role of inflammatory processes in the pathology of AMD.

Ion mobility-based LC-MS/MS has been used to study differences in the protein content of the high-density lipoprotein (HDL) fraction isolated from BrM-enriched tissues vs. plasma in the same individuals [54]. The results showed a striking over-representation of Apolipoprotein B (APOB) and Apolipoprotein E (APOE) in BrM. Since these isoforms bind to glycosaminoglycans, the authors proposed that the deposition of these lipoproteins may play a role in the downstream effects that contribute to RPE dysfunction and destruction, characteristic of AMD. To test whether APOE and APOB could be therapeutic targets for AMD, the anti-inflammatory 5A apolipoprotein A-1 (APOA1) mimetic peptide was used in a mouse model of AMD. The 5A peptide was able to modulate the proteomic profile of circulating HDL and prevent some of the potentially harmful changes in protein composition resulting from the high-fat, high-cholesterol diet in this model.

### 3.4. Proteomics on Drusen in AMD

Drusen are extracellular deposits, composed mainly of lipids, polysaccharides, proteins and glycosaminoglycans, that accumulate between the basal side of the RPE and the BrM, and are considered as risk factors for the development of AMD [105]. From a clinical point of view, drusen are classified into different types based on their relative size, shape, imaging characteristics and location.

The protein composition of drusen isolated from eye dissections from AMD patients and controls has been studied using LC-MS/MS [49]. Some proteins such as as vitronectin, TIMP3 or clusterin were common to both groups, while others such as crystallins were more frequently detected in the disease group. Furthermore, immunoblot analysis showed a higher level of crosslinked species and carboxyethyl pyrrole (CEP) adducts in drusen from patients, which reinforces the importance of oxidative processes in the pathogenesis of AMD. In another study, exosome markers CD63 and LAMP2 were detected in drusen from the eyes of AMD donors but not in age-matched controls. Interestingly, CD63 co-localized in these samples with other proteins characteristic of drusen, such us amyloid β, α-B-crystallin, C5b-9 and CFH, suggesting that the release of intracellular proteins via exosomes by the aged RPE may contribute to the formation of drusen [51].

### 3.5. Proteomics on Vitreous Humor in AMD

Vitreous humor is a colorless, transparent gelatinous substance filling the vitreous cavity, the region between the lens and the retina in the posterior segment of the eye [106]. It is surrounded by a collagen layer called the vitreous membrane. In addition to helping to maintain the normal shape of the ocular globe, it also acts as a reservoir of metabolites for the surrounding tissues and as a barrier to avoid the diffusion of substances between the retina and the anterior segment [107]. Since the vitreous humor is in direct contact with the lens, retina, macula and retinal vessels, the vitreous is, a priori, a promising source of biomarkers for the study of AMD and other ocular pathologies [108]. Furthermore, the vitreous fluid is the target in which intravitreal anti-VEGF injections, the gold standard treatment for nAMD, exert their therapeutic action [78]. However, to date, very few human-based studies on biomarker discovery in vitreous fluid have been published due to the difficulty of sample collection from living specimens [109].

Koss et al. used capillary electrophoresis coupled to mass spectrometry (CE-MS) and identified a set of 19 proteins accumulated in the vitreous fluid of AMD patients, most of which were related to acute-phase response and blood coagulation [59]. Among them, Alpha-1-antitrypsin was orthogonally validated in an independent set of AMD patients using Western blot analysis. In a subsequent study, the same group used a combination of CE-MS and LC-MS approaches to identify four potential biomarkers of nAMD progression in the vitreous fluid of patients with different degrees of CNV [60]. Validation using ELISA showed the best results for clusterin and PEDF. Clusterin has been related to cytoprotective effect in the retina, reducing apoptosis and ROS levels [110]. It has been hypothesized that clusterin can contribute to AMD pathogenesis through its potential role in modulating the complement system [111], including some of the components with genetic variants considered as risk factors for AMD, such as C3 and CFH [13].

More recently, Schori and colleagues [61] used label-free LC_MS/MS to establish the proteomic landscape in the vitreous of patients with dry AMD, nAMD and diabetic retinal disease (PDR). They identified different clusters of upregulated proteins for each patient group. Interestingly, complement and coagulation cascade appeared to be specially highly modulated in PDR, whereas the alteration of oxidative stress and focal adhesion pathways were characteristic of dry AMD and nAMD, respectively.

### 3.6. Proteomics on Aqueous Humor in AMD

The aqueous humor (AH) is a clear liquid that occupies the anterior and posterior chambers of the eye. Its composition is similar to that of plasma, although the protein concentration is much lower. It also contains electrolytes and ascorbate [112]. AH maintains intraocular pressure, provides nutrients and oxygen to the surrounding eye tissues lacking blood vessels and also removes their waste products [107].

A DIA quantitative proteomics study has recently been conducted in patients receiving anti-VEGF therapy [68]. Increased APOB100 levels were detected in pro re nata (PRN)-treated patients who required less frequent injections. Of interest, APOB100 accumulates within Bruch’s membrane as an early component of drusen [113]. Furthermore, APOB100 expression was higher in AMD eyes compared with healthy controls but was lower in eyes developing CNV, consistent with the protective role that has been attributed to this protein. A DIA-based approach was also used by Baek et al. [62] to study the proteome of the AH of dry AMD presenting soft drusen and/or reticular pseudodrusen. Eight proteins, APOA1, CFHR2 and CLUS among them, were previously described as major components or regulators of drusen. An additional set of three proteins (SERPINA4 protein, lumican, and keratocan) with no previous link with drusen formation were also increased in AH from dry AMD patients. Specifically, lumican and keratocan are involved in keratan sulphate proteoglycan (PG) biosynthesis and ECM remodeling, which could be partially linked to the ECM degradation that occurs in BrM during AMD development [102].

As previously described for vitreous humor, aqueous clusterin has recently been proposed as a biomarker for AMD progression by Rinsky et al. [66]. Clusterin was first detected as being overrepresented in the aqueous humor of nAMD patients vs. controls (n = 10 in both cases), which was later validated using ELISA in a larger cohort including nAMD patients (n = 15) and aAMD patients (n = 15) and controls (n = 20).

In a pilot study recently conducted by Coronado et al. [64], a proteomic analysis of the AH was conducted to gain a deeper understanding of the molecular pathways leading to choroidal neo-angiogenesis. A small cohort of 15 patients was divided into three groups; those with nAMD who demonstrated a good response to anti-VEGF intravitreal injections during follow-up, those with anti-VEGF-resistant nAMD who demonstrated choroidal neovascularization activity during follow-up, and control patients without systemic diseases or signs of retinopathy. Among the 185 discriminatory proteins, 39 were selected as potential disease effectors, including players of lipid metabolism (RBP3, APOA1), oxidative stress (SOD, GPX3), the complement system (C3, C7), inflammatory pathways (KLKB1, PEDF) and angiogenesis (TIMP1, VEGFR-1). Specifically, VEGFR-1 was up-regulated in non-responsive patients. According to the authors, this finding could explain the pathological tolerance that some patients develop to the gold-standard treatment for AMD and the persistence of the disease.

Exosomes isolated from AH collected from 28 AMD and 25 control eyes were lysed and the protein extracted for subsequent DDA label-free LC-MS/MS analysis by Tsai et al. [70]. Interestingly, gene ontology analysis showed that the only gene set enriched in AMD vs. the control was the lipoprotein metabolic process. APOA1, clusterin, C3 and opticin were among the proteins that significantly accumulated in AMD. Furthermore, AH at different time points was collected from only two AMD patients who received continuous anti-VEGF injections of ranibizumab every 12 weeks. LC-MS/MS analysis showed a progressive decrease in SERPINA1 and AZGP1 proteins in both patients. Since SERPINA1 promotes cell migration [114] and AZGP1 could enhance cell proliferation and the epithelial–mesenchymal transition (EMT) [115], the authors propose these two proteins as biomarkers for the therapeutic effect of anti-VEGF therapy in AMD.

Cytokine levels were measured, using multiplex antibody-based arrays, in the AH of nAMD patients and controls [65]. The CNV type was determined via the fluorescein angiography (FA) pattern. Several members of the C-C motif chemokine family (CCLs 2, 3 and 4) and VEGF were significantly increased in the nAMD group vs. the control group. When the two nAMD groups were compared separately vs. the control group, VEGF was found to be specifically increased in type 2 or classic CNV, characterized by increased neovascularization in the subretina and worse disease prognosis than patients with type 1 CNV [116]. Based on these results, the authors suggested that, in patients with type 1 CNV, also known as “occult” CNV, treatment based on VEGF inhibition alone may not be sufficient to achieve clinical benefits.

### 3.7. Proteomics on Tear Fluid in AMD

Tear fluid provides a non-invasive and easy source for sensitive proteomics to detect putative biomarkers of ocular surface health [117]. It is produced by lacrimal and accessory glands, as well as by meibomian glands and goblet cells, and is mainly composed of lipids, water and mucin [42]. Tear film is usually collected from the eye onto a Schirmer strip, although there are other alternatives such us the use of glass capillaries [118]. When deciding on the proper approach, it is important to consider that stimulated and non-stimulated tear film do not share all of the same biochemical properties. It is accepted that the use Schirmer strips triggers more intense tearing, which is helpful for better sample collection, but in turn, leads to underestimation of the actual protein concentration [43].

Historically, lactoferrin (LF), IgE and MMP-9 have been the most common translational biomarkers studied in tear film and their usefulness has been validated in dry eye disease [119], allergic conjunctivitis [120], keratoconus [121] and inflammatory conjunctivitis [122,123]. As for nAMD, the use of two-dimensional electrophoresis followed by the MALDI-TOF/TOF mass spectrometry approach in a recent study [67] has led to the identification of a set of dysregulated tear film proteins including several proteins related to inflammation and neovascularization, such as allograft inflammatory factor 1 (AIF1), ATP-dependent translocase (ABCB1) and annexin-1. A previous study from the same group included patients with both “wet” and “dry” AMD, as well as control individuals [63].

To investigate the role of altered metal homeostasis in AMD, a targeted ELISA-based analysis was recently used to measure the levels of a panel of metal-binding proteins of interest in the tear film of 60 patients, including 31 individuals diagnosed with the GA-AMD form [71]. The protein panel consisted of LF, S100 calcium-binding protein A6 (S100A6), metallothionein 1A (MT1A), CFH, clusterin and amyloid precursor protein (APP). The results indicated upregulation of MT1A and S100A6 in GA-AMD patients. The work was complemented by a multi-elemental analysis of the levels of Ca, Mg, P, Na, Zn, Fe and Cu using Inductively Coupled Plasma Mass Spectrometry (ICP-MS). Multivariate logistic regression and machine learning models were applied, and the panel consisting of MT1A, Na and Mg was found to predict AMD disease in 73% of cases. As a conclusion, the authors proposed a role for metal homeostasis in the progression of AMD.

Additionally, recently, VEGF levels in tears and serum were simultaneously measured in the same cohort of patients [69]. The cohort was composed of 108 individuals split into three categories (early AMD, late AMD and controls), each of which contained a third of the patients. The main conclusion of this work was that the tear level of VEGF presented high sensitivity and specificity as a predictor of the severity of the disease. On the contrary, the serum level of VEGF was found to be non-specific and non-predictive. Interestingly, the analysis of the demographic characteristics showed significant differences between controls and late AMD individuals in lifestyle variables, specifically cigarette smoking and alcohol consumption.

### 3.8. Proteomics on Blood in AMD

Novel Aptamer-Based proteomic technologies have been applied to the study of AMD biomarkers in plasma. This approach was used to differentiate the proteomic plasma signature of GA and nAMD patients vs. cataract controls [41]. Vinculin levels were significantly higher in nAMD patients, a result that was in concordance with previous mass spectrometry-based studies by Kim et al. [79]. Vinculin is a well-known regulator of apoptosis with additional roles in cell growth, migration, differentiation and survival. On the other hand, the same group validated their LC-MS/MS results via ELISA in two different cohorts of patients including healthy controls and both early AMD and exudative AMD patients [80]. The results showed that two proteins related to inflammation, PLTP and MASP-1, could be useful as candidate biomarkers for AMD progression. ROC and multivariate regression analysis indicated excellent diagnostic accuracy, especially for PLTP. Using the same technology, the proteogenomic signature of AMD in blood has recently been investigated [84] in the “Age, Gene/Environment Susceptibility Reykjavik Study” (AGES-RS) cohort [124]. The authors defined a set of 28 AMD-associated serum proteins. Subsets of these were specifically linked to the distinct stages of the disease and some could be useful to predict disease progression. For instance, serum levels of PRMT3, an arginine methyltransferase controlling ribosomal activity [125], were elevated in early AMD patients who subsequently progressed to GA, but not in those who progressed to nAMD.

Other protein biomarkers have been studied at the systemic level (Figure 3). Given that VEGF is the primary therapeutic target in nAMD, elevated levels of this molecule in the blood of patients could be, a priori, expected. However, there have been large discrepancies across different studies so far. VEGF was found to be increased in blood in the studies by Lip et al. [72] and Tsai et al. [74], but on the contrary, neither Carneiro et al. [77] nor Gu et al. [78] found significant differences between patients and controls. Inconsistent results have been also found for von Willebrand factor. This factor is released when endothelial cells are damaged, and it has been proposed as an indicator of endothelial damage or dysfunction in subjects with AMD [126]. Whereas one study showed higher levels in AMD compared with controls [72], more recent studies found no such association [75,76].

Regarding cholesterol transport and metabolism, more controversy has been added when studying the role of APOE polymorphisms in the development of AMD. APOE ϵ4 and ϵ2 isoforms decrease and increase the risks, respectively, for AMD [127,128,129]. However, APOE absence in humans and mice does not significantly affect the retina [130], indicating the existence of compensatory mechanisms that minimize the retinal impact of this absence.

A role in the systemic inflammatory processes associated with the development of iAMD has been proposed for plasma C-C chemokines. C-C chemokines are soluble mediators of inflammation-related chemotaxis and features of AMD, including drusen-like structures at the level of the RPE. CCL2 concentrations have been reported to be increased in patients with iAMD compared with controls, whereas CCL3 and CCL5 have been significantly decreased [82]. Additionally, RPE disruption and photoreceptor degeneration has previously been observed in CCL2 deficient mice [131].

The strong genetic association of the variants of Complement factor H (CFH) with AMD has also been explored from a proteomic point of view. A targeted, selected reaction monitoring (SRM) assay was developed by Zhang et al. to reliably quantify the Y402H and I62V variants [81], a challenging task due to the single amino acid substitutions and high sequence homology between complement factor H and complement factor H-related proteins.

### 3.9. Proteomics on Urine in AMD

Several studies have identified common pathogenetic mechanisms underlying renal and retinal diseases [132,133]. Interestingly, the vascular networks of the glomerulus and choroid present similar structures, and the renin–angiotensin–aldosterone hormonal cascade is found in both the kidney and the eye [134]. Increased levels of urinary markers of oxidative stress such us F2-isoprostanes, a marker of lipid peroxidation, and cadmium have been associated with the progression of AMD [135,136]. Chronic Kidney Disease (CKD) and the main ocular diseases (AMD, diabetic retinopathy, glaucoma and cataract) share common vascular risk factors including diabetes, hypertension, smoking and obesity, as excellently reviewed by Wong et al. [137].

Based on all this, urine has also been used, in recent years, as a non-invasive easy-to-collect source for biomarker discovery, with the aim of identifying not only the metabolomic [138,139], but also the proteomic signature of the different sub-types of AMD. A tandem mass tagged (TMT) approach identified panels of proteins characteristic of eAMD, GA and nAMD [83]. ELISA validation of some of the candidates showed that SERPINA-1, TIMP-1 and APOA1 were significantly over-expressed in AMD vs. controls.

### 3.10. Therapeutic Challenges and Future Directions

Proteomics-based biomarker discovery for AMD development and progression has identified a set of diverse modulated proteins, summarized in Table 4. The current available therapies, focused on targeting VEGF or inflammation, are effective approaches, but only in neovascular AMD. Thus far, translational research in this field has been strongly limited by the difficulties in establishing good experimental models [140], due to anatomical differences of the structure of the eye between rodents and humans [141] or failure to recapitulate the multifactorial characteristics of the disease [142].

Targeting the complement cascade appears to be the more promising therapeutic approach, as has been recently comprehensively reviewed by Patel and colleagues [143] but no drug has been marketed yet.

Currently, ongoing phase III trials include complement factors such as C3 (APL-2, pegcetacoplan) [144,145] or C5 (avacincaptad pegol) [146] as therapeutic targets with promising results. Specifically, therapeutic targeting of C3 using APL-2, a peptide inhibitor that is administered intravitreally, has very recently been shown to be effective even earlier in the progression of AMD prior to the development of GA [147]. On the contrary, a finished phase III trial targeting complement factor D (lampalizumab) showed no difference in the progression of GA compared with a placebo [148].

Finally, innovative solutions for ocular delivery based on hydrogels [149], nanocarriers [150] or polymeric micelles [151] will be of paramount importance for maximizing bench-to-bedside transition and to improve patient adherence to the new therapeutic drugs.

## 4. Conclusions

AMD is a prevalent condition representing the leading cause of irreversible visual impairment in Western countries in the elderly population. Although it is accepted that the activation of a cascade of proinflammatory and proangiogenic factors, driven by damage to the choriocapillaris, the RPE and the outer retina, plays a key role in the development of the disease, the exact pathogenic mechanisms shared by the different forms of AMD remain elusive and need to be elucidated to therapeutically address the early stages of the disease. Proteomics has given us, in the last half decade, new clues that will help us in this endeavor. Examples of this are the involvement of detoxification pathways, the regulation of the complement by clusterin, the involvement of several members of the C-C motif chemokine family, the role of EVs in the formation of drusen, and the molecular control of processes such us ECM remodeling and EMT as triggering factors for AMD. We strongly believe that proteomics will be, in the coming years, a fundamental tool to elucidate the precise molecular role of these candidates and to study the clinical progression of patients.

## Figures and Tables

**Figure 1 ijms-23-14759-f001:**
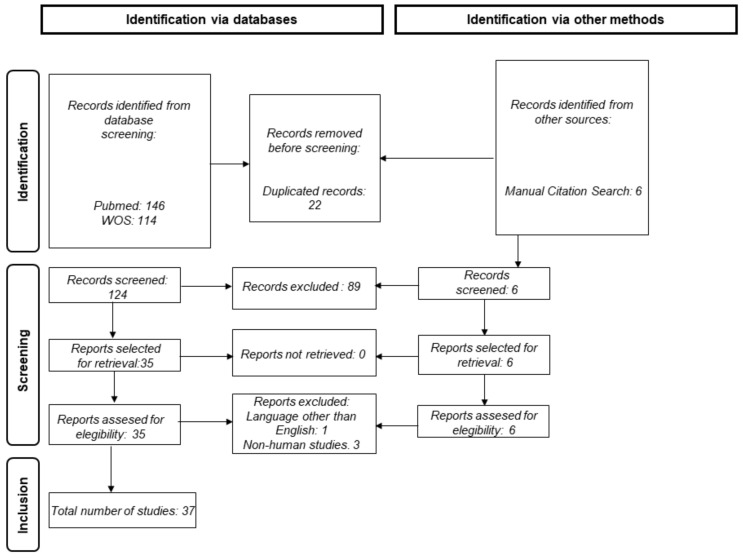
PRISMA workflow followed for the systematic revision of the literature.

**Figure 2 ijms-23-14759-f002:**
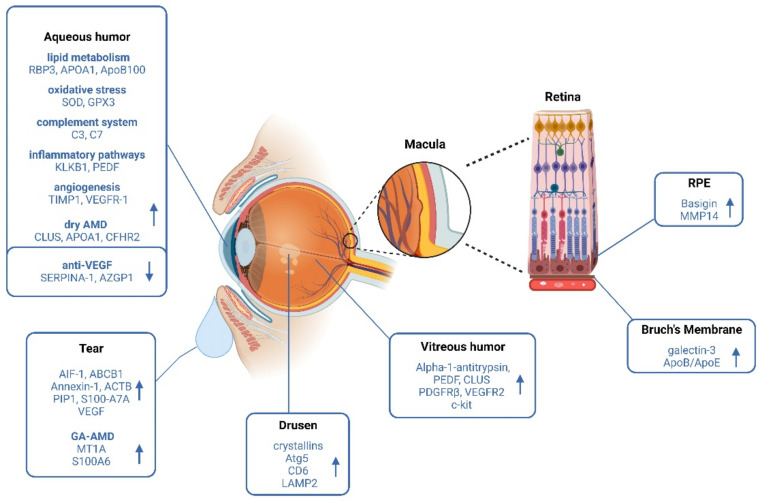
Schematic representation of the main findings regarding biomarker discovery in AMD (proteins increased: up-arrow; proteins decreased: down-arrow) using ocular tissues/fluids as a source. Created using BioRender.com (accessed on 5 October 2022).

**Figure 3 ijms-23-14759-f003:**
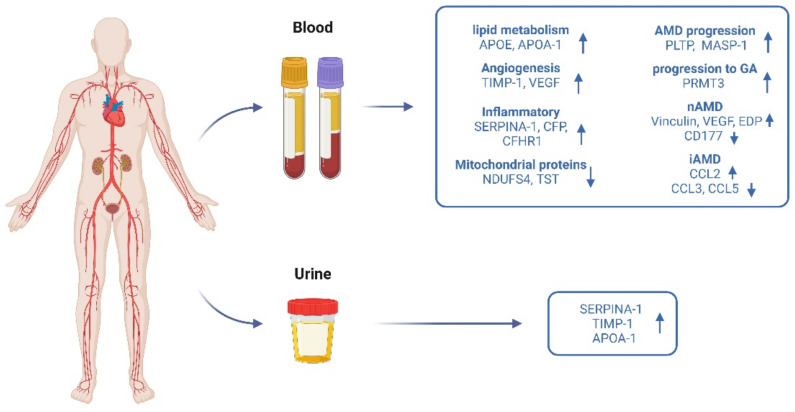
Schematic representation of the main findings regarding biomarker discovery in AMD (proteins increased: up-arrow; proteins decreased: down-arrow) using systemic fluids as a source. Created using BioRender.com (accessed on 5 October 2022).

**Table 1 ijms-23-14759-t001:** Proteomic studies on cells/ocular tissues in AMD.

Study	Biomarker Source	Characteristics of the Cohort	Proteomic Approach(es)	Main Findings
Crabb et al., 2002 [49]	Drusen and BrM	18 controls 5 donors with AMD	Label-free LC-MS/MS	A total of 129 proteins identified. Crystallins are more frequently detected in the diseased group.
Alcazar et al., 2009 [50]	Exosomes from Hydroquinone-stimulated ARPE-19 cells	N.A.	SDS-PAGE coupled to LC-MS/MS Immunofluorescence	Proteins involved in oxidative phosphorylation, cell junction, focal adhesion, cytoskeleton regulation and immunogenic processes. Basigin and MMP14 could be involved in progression of dry AMD.
Wang et al., 2009 [51]	RPE tissue, drusen and ARPE-19 cells	12 eyes (six donors) with no history of AMD 4 eyes (2 donors) with history of AMD 8 eyes (8 donors) documented AMD	Immunoblot, ELISA and Luminex	Drusen in AMD donor eyes contain markers for autophagy (atg5) and exosomes (CD63 and LAMP2). Exosome markers are characteristic of drusen from AMD patients and co-localize in the RPE/choroid complex.
Yuan et al., 2010 [52]	Bruch’s membrane	10 early/mid-stage dry AMD 6 advanced dry AMD, 8 wet AMD 25 normal control post-mortem eyes	iTRAQ (isobaric labeling DDA- LC-MS/MS)	Retinoid-processing proteins increased in early/mid dry AMD. Galectin-3 increased in advanced dry AMD.
Biasutto et al., 2013 [53]	Exosomes from ARPE-19 under oxidative stress conditions	N.A.	Reverse-phase assay	Identification of a subset of phosphorylated proteins including PDGFRβ, VEGFR2 and c-kit that are also detected in the vitreous of AMD patients.
Kelly et al., 2020 [54]	Bruch´s membrane	3 donors with AMD	Ion mobility-based LC-MS/MS	APOE and APOB over-represented in HDL from BrM vs. plasma.
Flores-Bellver et al., 2021 [55]	RPE monolayers generated from induced pluripotent stem cells (iPSCs) derived of CD34+ cord blood mesenchymal stem cells	N.A.	Label-free LC-MS/MS ELISA Immunoblot	Drusen-associated proteins exhibit distinctive directional secretion mode altered in AMD pathological conditions (e.g., chronic exposure to cigarette smoke).
Cai et al., 2022 [56]	RPE cells from donor´s eyes	4 donors with AMD high-risk alleles 2 donors with AMD low-risk alleles	iTRAQ (isobaric labeling DDA- LC-MS/MS)	Exposure of high-risk donor-derived RPE cells to the serum from smokers enhances molecular pathways related to development of AMD.
Senabouth et al., 2022 [57]	iPSCs generated from skin fibroblasts	43 GA 36 Controls	TMT (isobaric labeling DDA- LC-MS/MS)	GA patients present mitochondrial dysregulation characterized by an increase in Complex I levels and activity.
Zauhar et al., 2022 [58]	RPE and choroid fibroblasts, pericytesand endothelial cells	N.A.	Label-free LC-MS/MS	Classical complement pathway involvement more robust in retina. New cellular targets for therapies directed at complement.

**Table 2 ijms-23-14759-t002:** Proteomic studies on ocular fluids in AMD.

Study	Biomarker Source	Characteristics of the Cohort	Proteomic Approach(es)	Main Findings
Koss et al., 2014 [59]	Vitreous humor	73 naïve patients 15 control samples from patients with idiopathic floaters	CE-MS	Acute-phase response and blood coagulation up-regulated in AMD, Alpha-1-antitrypsin among them.
Nobl. et al., 2016 [60]	Vitreous humor	128 nAMD 24 controls	CE-MSELISA	Clusterin and PEDF levels are predictive for nAMD.
Schori et al., 2018 [61]	Vitreous humor	6 patients with dry AMD 10 patients with nAMD 9 patients with proliferative diabetic retinopathy 9 patients with epiretinal membrane	Label-free LC-MS/MS	Oxidative stress and focal adhesion pathways modulated in dry AMD and nAMD, respectively.
Baek et al., 2018 [62]	Aqueous humor	13 patients with cataract, 11 patients with dry AMD and 2 patients with no retinal diseases	DIA-MS (SWATH)ELISA	A total of 8 proteins involved in drusen development, including APOA1, CFHR2 and CLUS, are accumulated in the AH of dry AMD patients.
Winiarczyk et al., 2018 [63]	Tear	8 wet AMD, 6 dry AMD and 8 controls	2D-LC-MALDI-TOF	Graves disease carrier protein, actin cytoplasmic 1, prolactin-inducible protein 1 and protein S100-A7A are upregulated in the tear film samples isolated from AMD Patient.
Coronado et al., 2021 [64]	Aqueous humor	Group 1: nAMD patients: good responders to anti-VEGF) Group 2: nAMD patients (poorly/non-responsive to anti-VEGF) Group 3: patients without systemic diseases or signs of retinopathy	Label-free LC-MS/MS	A total of 39 potential disease effectors, including players of lipid metabolism, oxidative stress, inflammation and angiogenesis. VEGFR-1 is up-regulated in non-responsive patients, which could explain resistance to treatment.
Joo et al., 2021 [65]	Aqueous humor	13 nAMD patients (type 1: n = 8; type 2: n = 5) and 10 controls undergoing cataract surgery with no retinal diseases	Multiplexed antibody-based array	VEGF is specifically increased in nAMD patients with type 2 CNV.
Rinsky et al., 2021 [66]	Aqueous humor	Discovery: 10 nAMD patients and 10 controlsValidation: 20 controls, 15 atrophic AMD and 15 nAMD patients	Intensity-based label-free quantification (MS1) Multiplex ELISA	Clusterin overrepresented in the aqueous of nAMD patients.
Winiarczyk et al., 2021 [67]	Tear	15 nAMD patients 15 controls	2D-LC-MALDI-TOF	AIF-1, ABCB1 and annexin-1 are higher in AMD.
Cao et al., 2022 [68]	Aqueous humor	122 nAMD with anti-VEGF therapy	DIA-MS (SWATH)	APOB100 expression is higher in AMD vs. control.
Shahidatul-Adha et al., 2022 [69]	Tear and plasma	36 eAMD 36 lAMD 36 controls	ELISA	Tear VEGF level presents high sensitivity and specificity as a predictor of the severity of the disease.
Tsai et al., 2022 [70]	Exosomes from Aqueous humor	28 eyes from AMD patients (2 of them followed during Ranibizumab treatment). 25 control eyes from senile cataract patients without other ocular or systemic diseases	Label-free LC-MS/MS	APOA1, clusterin, C3 and opticin significantly accumulated in AMD. Anti-VEGF therapy progressively decreases levels of SERPINA1 and AZGP1.
Valencia et al., 2022 [71]	Tear	60-patient cohort: 31 with diagnosed GA-AMD	ELISA	Upregulation of MT1A and S100A6 in GA-AMD patients.

**Table 3 ijms-23-14759-t003:** Proteomic studies on systemic fluids in AMD.

Study	Biomarker Source	Characteristics of the Cohort Used for the Proteomic Study	Proteomic Approach(es)	Main Findings
Lip et al., 2001 [72]	Plasma	28 “dry” AMD 50 “exudative” AMD 25 “healthy” controls	ELISA	VEGF and VWF significantly increased in AMD.
Sivaprasad et al., 2005 [73]	Plasma	26 nAMD 30 eAMD 15 controls	ELISA	Elastin-derived peptides elevated in the serum of nAMD patients vs. eAMD and control subjects.
Tsai et al., 2006 [74]	Plasma	17 dry AMD 42 wet CNV/AMD 18 scar/AMD64 non-AMD	ELISA	VEGF significantly increased in CNV/AMD.
Wu et al., 2007 [75]	Serum	159 eAMD 38 lAMD 433 controls	ELISA	No consistent pattern of association found between AMD and circulating inflammatory markers.
Rudnicka et al., 2010 [76]	Serum	81 AMD 77 controls	ELISA	FVIIc and possibly F1.2 are inversely associated with the risk of AMD. No evidence of associations between AMD and systematic markers of arterial thrombosis.
Carneiro et al., 2012 [77]	Plasma	43 exudative AMD: 19 ITV ranibizumab 24 ITV bevacizumab 19 age-related controls	ELISA	No basal differences in VGEF between AMD and controls. Significant reduction in VEGF levels with intravitreal bevacizumab.
Gu et al., 2013 [78]	Serum	39 neovascular AMD with single-dose ranibizumab 39 healthy controls	ELISA	No basal differences in VGEF between AMD and controls. VEGF levels significantly decrease after injection but increase later.
Kim et al., 2014 [79]	Plasma	20 exudative AMD 20 healthy controls Validation: 233 case–control samples	LC-MS/MS ELISA WB	Vinculin is identified as a potential plasma biomarker for AMD.
Kim et al., 2016 [80]	Plasma	90 healthy controls 49 eAMD and 87 exudative AMD	ELISA	MASP1, and especially PLPT useful as predictors of AMD progression.
Zhang et al., 2017 [81]	Plasma	344 adults	Selected Reaction Monitoring	Development of a method to quantify Y402H and I62V AMD-associated variants of Complement Factor H.
Lynch et al., 2019 [41]	Plasma	10 nAMD 10 GA 10 age-matched cataract controls	Aptamer-based proteomics	Higher levels of vinculin and lower levels of CD177 are found in patients with neovascular AMD compared with controls.
Palestine et al., 2021 [82]	Plasma	210 iAMD 102 controls	Multiplex	CCL3 and CCL5 significantly decreased and CCL2 increased in patients with iAMD compared with controls.
Sivagurunathan et al., 2021 [83]	Plasma and urine	23 controls 61 AMD	Shotgun LC-MS/MS (TMT) ELISA	SERPINA-1, TIMP-1 and APOA-1 higher in AMD.
Emilsson et al., 2022 [84]	Serum	Discovery: 1054 eAMD 112 GA pure 160 nAMD 183 GA + nAMD Validation: 15 subjects for each category	Aptamer-based proteomics ELISA	Determination of a set of 28 AMD-associated proteins including CFHR1, TST, DLL3, ST6GALNAC1, CFP and NDUFS4. PRMT3 proposed as predictor for progression to GA.

**Table 4 ijms-23-14759-t004:** Main biomarkers of AMD development and progression.

Process	Protein Biomarkers	References
RPE redox maintenance	CCLs	[65,82,131]
	Crystallins	[49,51]
Regulation of neovascularization	VEGF	[10,65,69,72,74]
	VEGFR	[53,64]
	TIMP1	[64,83]
	Opticin	[70]
Metal homeostasis and ECM remodeling	S100A6	[71]
	CFH, CFHR	[49,62,71,81]
	TIMP1, TIMP3	[64,83]
	Elastin	[73,103,104]
	MMP14	[50,100]
Lipoprotein metabolism	APOA1	[54,62,64,70,83]
	APOB	[54,68,113]
	Clusterin	[60,66,70,71,110,111]
Complement cascade	C3	[64,70]
	CFH, CFHR	[49,62,71,81]
	C5	[49,51]
	Clusterin	[60,66,70,71,110,111]

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
