# Peer review of "Recent Advances in Proteomics-Based Approaches to Studying Age-Related Macular Degeneration: A Systematic Review"

_ijms, 2022, doi:10.3390/ijms232314759_

Round 1

Reviewer 1 Report

A very good review on a subject that is not discussed enough.

Well done.

Author Response

RESPONSES TO REVIEWER 1:

A very good review on a subject that is not discussed enough.

Well done.

RESPONSE: Thank you very much for your comment, we hope the present manuscript will be of help to researchers in the field.

Reviewer 2 Report

Recent Advances in Proteomic-based Approaches to Study Age-Related Macular Degeneration: A Systematic Review

 Manuscript Summary:

The authors are systematically reviewing proteomic studies to identify biomarkers and disease mechanisms in age-related macular degeneration (AMD). The authors reviewed the previous literature about various proteomic studies to delineate the underlying molecular mechanism of AMD at different stages of the disease. From reviewing the previously published literature, the authors concluded the proteins of lipid metabolism and oxidative stress are mainly involved in the onset of AMD, while in the advanced stage of the disease, proteins of extracellular matrix integrity and angiogenesis play pivotal roles. The authors have presented a relevant detailed review of the literature and highlighted the significance of proteomic studies for addressing the disease mechanism of complex diseases like AMD. However, the following comments need to be addressed by the authors to enhance the review article:

Major comments:

1.    The authors need to cite the reference for the statements given in the introductory paragraph, lines no. 35-37.

2.    The authors need to cite literature for lines 62-63 that link ROS and AMD association. There are previous documented studies showing the association between ROS and AMD pathogenesis. Eg. Belleza 2018, Jarret et al, 2012

3.    The authors need to cite the reference at the end of the statement in line numbers 85-89. “The neovascularization process can be classified in three forms using optical coherence tomography (OCT): Type 1 CNV refers to vessels beneath the RPE, whereas type 2 CNV is characterized by vessels expanding into the subretinal space between the neurosensory retina and the RPE [31], and type 3 by retinal angiomatous proliferation (Ref).”

4.    In line no. 90, the authors have mentioned that recent studies have revealed that early AMD signs can be detected in patients under 30 years. But the authors cited only one reference for this statement.  

5.    In Table 2. the authors enlisted only two vitreous humor studies in AMD. The study performed by Schori et al., 2018 is not mentioned in the table.

6.    The authors may use the terms aqueous humor and vitreous humor in tables and text rather than using terms aqueous fluid and vitreous fluid respectively in the tables.

7.    The authors need to align the table in the chronological order of published year for the similar type of biological fluids.

8.    In line no. 172, the authors mentioned the top N. It is not clear what N indicates.

9.    In line no. 197- 198, the authors mentioned “Hence, for quantitative proteomics is generally necessary to quantify the samples, both at the level of total protein and subsequently at the level of peptide extracts”. It is not clear whether the authors mention about number of peptides or their intensity.

10. Please rewrite sentence 222-224, which is not clear.

11. No reference quoted for the statement written in line no. 228-230.

12. Reference for the statement “The extracellular vesicles (EVs) have been revealed as key players in biological processes such as aging, cell homeostasis and disease” is missing in lines 231-232.

13. Need to add citations in section 3.5, lines 317-325.

14. In line no. 372, the authors can use complement proteins as C3 and C7 instead of using CO3 and CO7.

15.  In Figure 2, the authors need to exactly label the regions, e.g. A line from the cornea is shown as aqueous humor, and the sclera is labelled as vitreous humor. Please label correctly.

16. The proteomic studies and genetic studies clearly demonstrated the involvement of the complement system, which has a strong association with AMD pathogenesis. But in the abstract authors mentioned only pathways associated with angiogenesis, ECM, and lipid metabolism as pivotal for AMD pathogenesis.

Minor comments:

-        Expand the abbreviations in the first use. eg. ApoB, ApoE

-        Need to add references wherever applicable.

Author Response

RESPONSES TO REVIEWER 2:

Manuscript Summary:

The authors are systematically reviewing proteomic studies to identify biomarkers and disease mechanisms in age-related macular degeneration (AMD). The authors reviewed the previous literature about various proteomic studies to delineate the underlying molecular mechanism of AMD at different stages of the disease. From reviewing the previously published literature, the authors concluded the proteins of lipid metabolism and oxidative stress are mainly involved in the onset of AMD, while in the advanced stage of the disease, proteins of extracellular matrix integrity and angiogenesis play pivotal roles. The authors have presented a relevant detailed review of the literature and highlighted the significance of proteomic studies for addressing the disease mechanism of complex diseases like AMD. However, the following comments need to be addressed by the authors to enhance the review article:

RESPONSE: We would like to thank reviewer 2 for her/his helpful comments and corrections.

Major comments:

  1. The authors need to cite the reference for the statements given in the introductory paragraph, lines no. 35-37.

RESPONSE: References included.

  1. The authors need to cite literature for lines 62-63 that link ROS and AMD association. There are previous documented studies showing the association between ROS and AMD pathogenesis. Eg. Belleza 2018, Jarret et al, 2012

RESPONSE: The sentence has been properly referenced.

  1. The authors need to cite the reference at the end of the statement in line numbers 85-89. “The neovascularization process can be classified in three forms using optical coherence tomography (OCT): Type 1 CNV refers to vessels beneath the RPE, whereas type 2 CNV is characterized by vessels expanding into the subretinal space between the neurosensory retina and the RPE [31], and type 3 by retinal angiomatous proliferation (Ref).”

RESPONSE: Reference included (Gigon et al., 2022).

  1. In line no. 90, the authors have mentioned that recent studies have revealed that early AMD signs can be detected in patients under 30 years. But the authors cited only one reference for this statement.  

RESPONSE: The reviewer is true. This is a very recent clinical observation which is documented, to our knowledge, only in one reference. We have changed the paragraph accordingly.

  1. In Table 2. the authors enlisted only two vitreous humor studies in AMD. The study performed by Schori et al., 2018 is not mentioned in the table.

RESPONSE: We thank the reviewer for pointing this missing piece of information that is now included in the manuscript.

  1. The authors may use the terms aqueous humor and vitreous humor in tables and text rather than using terms aqueous fluid and vitreous fluid respectively in the tables.

RESPONSE: We have homogenized the terms.

  1. The authors need to align the table in the chronological order of published year for the similar type of biological fluids.

RESPONSE: We have followed reviewer´s recommendations.

  1. In line no. 172, the authors mentioned the top N. It is not clear what N indicates.

RESPONSE: The sentence has been clarified in the new version of the manuscript.

  1. In line no. 197- 198, the authors mentioned “Hence, for quantitative proteomics is generally necessary to quantify the samples, both at the level of total protein and subsequently at the level of peptide extracts”. It is not clear whether the authors mention about number of peptides or their intensity.

RESPONSE: The sentence has been clarified in the new version of the manuscript.

  1. Please rewrite sentence 222-224, which is not clear.

RESPONSE: The sentence has been clarified in the new version of the manuscript.

  1. No reference quoted for the statement written in line no. 228-230.

RESPONSE: The reference for this statement is Senabouth et al which is mentioned in the previous sentence.

  1. Reference for the statement “The extracellular vesicles (EVs) have been revealed as key players in biological processes such as aging, cell homeostasis and disease” is missing in lines 231-232.

RESPONSE: The statement has been referenced.

  1. Need to add citations in section 3.5, lines 317-325.

RESPONSE: References for the paragraph has been completed with Gu et al., 2015 Tamhane et al., 2019 and Tamai et al., 1991.

  1. In line no. 372, the authors can use complement proteins as C3 and C7 instead of using CO3 and CO7.

RESPONSE: This has been corrected in this new version of the manuscript

  1. In Figure 2, the authors need to exactly label the regions, e.g. A line from the cornea is shown as aqueous humor, and the sclera is labelled as vitreous humor. Please label correctly.

RESPONSE: This has been corrected in this new version of the manuscript

  1. The proteomic studies and genetic studies clearly demonstrated the involvement of the complement system, which has a strong association with AMD pathogenesis. But in the abstract authors mentioned only pathways associated with angiogenesis, ECM, and lipid metabolism as pivotal for AMD pathogenesis.

RESPONSE: Thank you for bringing this into our attention, this has been also solved. Additionally, we have added a last subsection (3.10.) to discuss therapeutical targeting of complement cascade.

Minor comments:

-        Expand the abbreviations in the first use. eg. ApoB, ApoE

RESPONSE: This has been corrected in this new version of the manuscript.

-       Need to add references wherever applicable.

RESPONSE: This has been corrected in this new version of the manuscript.

Reviewer 3 Report

This is a good review that provides a brief history of the development of proteomics and a summary of proteomics researches for AMD to date. The following points could be modified.

Major points

1. The manuscript summarizes previous proteomics studies on RPE cells, Bruch's membrane, vitreous fluid, tear fluid, blood, and urine samples. However, the results obtained from each study were extremely diverse, and it does not appear that a protein that can uniquely explain the pathogenesis of AMD has been detected. Explain the reason for this variation in reporting among researchers.

2. It would be easier for the reader to understand if the proteins briefly summarized in L. 515-519 and those predicted to be involved in the pathogenesis of AMD in other representative proteomics studies are extracted and summarized in a new table.

Minor points

1. L. 51: "etologic" should be "etiologic.

2. L. 138: "AND" should be "AMD". 

3. Fig. 2: Drusen points to the wrong place. Correct.

4. Fig. 2: Vitreous humor points to the wrong place. Correct. 

5. L. 244: "component components" should be "components".

Author Response

RESPONSES TO REVIEWER 3:

This is a good review that provides a brief history of the development of proteomics and a summary of proteomics researches for AMD to date. The following points could be modified.

RESPONSE: We would like to thank Reviewer 3 for her/his helpful suggestions and corrections.

Major points

  1. The manuscript summarizes previous proteomics studies on RPE cells, Bruch's membrane, vitreous fluid, tear fluid, blood, and urine samples. However, the results obtained from each study were extremely diverse, and it does not appear that a protein that can uniquely explain the pathogenesis of AMD has been detected. Explain the reason for this variation in reporting among researchers.

RESPONSE: This important point is further discussed in new sub-section 3.10. and summarized in table 4.

  1. It would be easier for the reader to understand if the proteins briefly summarized in L. 515-519 and those predicted to be involved in the pathogenesis of AMD in other representative proteomics studies are extracted and summarized in a new table.

RESPONSE: Following Reviewer 3 recommendations we have included a new Table 4.

Minor points

  1. L. 51: "etologic" should be "etiologic.

RESPONSE: This has been corrected.

  1. L. 138: "AND" should be "AMD". 

RESPONSE: In this case “AND” refers to “proteomics” and “age-related macular degeneration”

  1. Fig. 2: Drusen points to the wrong place. Correct.

RESPONSE: Thank you for bringing this into our attention, this has been corrected.

  1. Fig. 2: Vitreous humor points to the wrong place. Correct. 

RESPONSE: Thank you for bringing this into our attention, this has been corrected.

  1. L. 244: "component components" should be "components".

RESPONSE: The sentence has been modified.

Reviewer 4 Report

Garcia-Quintanilla and colleagues provide an overview of proteomic research in age-related macular degeneration, delving into studies exploring research in ocular tissues and both ocular and systemic fluids. The review balances breadth and depth and includes many recent up-to-date articles. Overall, this is a well-written article yet can be improved further.

General Comments:

- The introduction is very long and can be highly distracting to the reader. The molecular level paragraph can be significantly shortened by removing detailed pathogenesis pathways and combining it with the key importance paragraph (lines 90-96), and merging the first 3 paragraphs by concisely mentioning the disease, prevalence, brief classification, and treatment in a few sentences. The proteomics paragraph can also be shortened to briefly mention the field and merge it with the paragraph starting at line 112. This is because the authors already provide a highly detailed review of proteomics in health and disease in section 3.1.

- Figures and tables are very well-made and greatly clarify the manuscript.

- The authors need to double check all abbreviations are listed in its extent before it is used throughout the manuscript (eg. VEGF in line 47 is the first time it is used but is not provided in its full form and then VEGF in parentheses.)

-Authors mention the impact of the complement factor system in the pathogenesis of dry AMD. It would be important for this review to discuss new therapies on the horizon that target the complement pathway (eg. pegcetacoplan) to provide a translational approach to the paper.

- A final sentence can be added to the conclusion section highlighting the importance of proteomics in therapeutic discovery and other potential themes the authors may want to add, as it remains incomplete to the reader in its current state.

Minor Comments:

- Lines 136-37: Can remove the End Note sentence as it does not add to the review.

- Capitalize the "F" in figures throughout the text.

Author Response

RESPONSES TO REVIEWER 4:

Garcia-Quintanilla and colleagues provide an overview of proteomic research in age-related macular degeneration, delving into studies exploring research in ocular tissues and both ocular and systemic fluids. The review balances breadth and depth and includes many recent up-to-date articles. Overall, this is a well-written article yet can be improved further.

RESPONSE: We would like to thank Reviewer 4 for her/his helpful comments and suggestions.

General Comments:

- The introduction is very long and can be highly distracting to the reader. The molecular level paragraph can be significantly shortened by removing detailed pathogenesis pathways and combining it with the key importance paragraph (lines 90-96), and merging the first 3 paragraphs by concisely mentioning the disease, prevalence, brief classification, and treatment in a few sentences. The proteomics paragraph can also be shortened to briefly mention the field and merge it with the paragraph starting at line 112. This is because the authors already provide a highly detailed review of proteomics in health and disease in section 3.1.

RESPONSE: We have followed Reviewer 4 recommendations in the introduction of the new version of the manuscript and we have shortened it. However, we think that the introduction has still to be strong enough, in terms of pathogenesis and molecular mechanisms of AMD for readers that are experts on proteomics but not on ocular inflammatory diseases.  We have tried our best to achieve this balance. 

- Figures and tables are very well-made and greatly clarify the manuscript.

RESPONSE: Thank you very much for your comment, figures have been re-checked and slightly improved in the new version of the manuscript.

- The authors need to double check all abbreviations are listed in its extent before it is used throughout the manuscript (eg. VEGF in line 47 is the first time it is used but is not provided in its full form and then VEGF in parentheses.)

RESPONSE: Thank you for pointing this, we have checked all the abbreviations in this new version.

-Authors mention the impact of the complement factor system in the pathogenesis of dry AMD. It would be important for this review to discuss new therapies on the horizon that target the complement pathway (eg. pegcetacoplan) to provide a translational approach to the paper.

RESPONSE: We have added a new subsection titled “therapeutical challenges and future directions” to discuss this important point and others related to therapeutics.

- A final sentence can be added to the conclusion section highlighting the importance of proteomics in therapeutic discovery and other potential themes the authors may want to add, as it remains incomplete to the reader in its current state.

RESPONSE: The conclusion is now finished with a highlighting sentence.

 Minor Comments:

- Lines 136-37: Can remove the End Note sentence as it does not add to the review.

 RESPONSE: This sentence has been removed.

- Capitalize the "F" in figures throughout the text.

RESPONSE: This has been corrected in the new version of the manuscript.

Round 2

Reviewer 3 Report

I found that the revised manuscript has been well improved enough to be published.